# Nephritis-Associated Plasmin Receptor (NAPlr): An Essential Inducer of C3-Dominant Glomerular Injury and a Potential Key Diagnostic Biomarker of Infection-Related Glomerulonephritis (IRGN)

**DOI:** 10.3390/ijms23179974

**Published:** 2022-09-01

**Authors:** Nobuyuki Yoshizawa, Muneharu Yamada, Masayuki Fujino, Takashi Oda

**Affiliations:** 1Division of Nephrology, Showanomori Hospital, Tokyo 196-0024, Japan; 2Department of Nephrology and Blood Purification, Kidney Disease Center, Tokyo Medical University Hachioji Medical Center, Tokyo 193-0998, Japan; 3National Institute of Infectious Disease, Tokyo 162-8640, Japan

**Keywords:** poststreptococcal acute glomerulonephritis (PSAGN), nephritis-associated plasmin receptor (NAPlr), immune complex-dominant glomerular injury, C3-dominant glomerular injury, NAPlr (Bacterial GAPDH) mediated glomerulonephritis, infection-related glomerulonephritis (IRGN)

## Abstract

Nephritis-associated plasmin receptor (NAPlr) was originally isolated from the cytoplasmic fraction of group A *Streptococci*, and was found to be the same molecule as streptococcal glyceraldehyde-3-phosphate dehydrogenase (GAPDH) and plasmin receptor (Plr) on the basis of nucleotide and amino acid sequence homology. Its main functions include GAPDH activity, plasmin-binding capacity, and direct activation of the complement alternative pathway (A-P). Plasmin trapped by deposited NAPlr triggers the degradation of extracellular matrix proteins, such as glomerular basement membranes and mesangial matrix, and the accumulation of macrophages and neutrophils, leading to the induction of plasmin-related endocapillary glomerular inflammation. Deposited NAPlr at glomerular endocapillary site directly activates the complement A-P, and the endocapillary release of complement-related anaphylatoxins, C3a and C5a, amplify the in situ endocapillary glomerular inflammation. Subsequently, circulating and in situ-formed immune complexes participate in the glomerular injury resulting in NAPlr-mediated glomerulonephritis. The disease framework of infection-related glomerulonephritis (IRGN) has been further expanded. GAPDH of various bacteria other than *Streptococci* have been found to react with anti-NAPlr antibodies and to possess plasmin-binding activities, allowing glomerular NAPlr and plasmin activity to be utilized as key biomarkers of IRGN.

## 1. Introduction

We have previously isolated and characterized a nephritogenic protein from group A *Streptococcus* (GAS) which we named the nephritis-associated plasmin receptor (NAPlr). This is homologous to the *Streptococcus* plasmin receptor (Plr) [1,2]. Evidence for the important roles of NAPlr and the related plasmin activity in the development of glomerulonephritis (GN) associated with streptococcal infection has been established.

The most popular theory regarding the pathogenic mechanism of post-streptococcal acute glomerulonephritis (PSAGN) is the immune complex (IC) theory, which involves glomerular deposition of the nephritogenic streptococcal antigen and the subsequent in situ formation and deposition of circulating ICs [3,4,5]. However, glomerular immunoglobulin deposition is not always prominent in this disease. In fact, 30% to 40% of patients with PSAGN are positive for C3 but negative for IgG in the glomeruli. Furthermore, the site of glomerular inflammation differs from that of IC deposition. The major site of inflammation is on the inner side of the glomerular tufts (endocapillary site), whereas the IC is localized on the outer side of the glomerular tufts (subepithelial site) in the early phase. Histological analysis revealed that NAPlr deposition and related plasmin activity are localized within the inner side of glomerular tufts where glomerular inflammation mainly occurs [6]. Therefore, we considered that NAPlr can directly induce in situ glomerular inflammation, independent of IC deposition in PSAGN patients.

Whereas most patients with PSAGN recover without any specific therapy, the prognosis of patients with infection-related glomerulonephritis (IRGN) is poor. Controlling the underlying infection and managing complications are essential for the effective treatment of IRGN. However, the prompt diagnosis of IRGN is often difficult as specific diagnostic biomarkers have not yet been identified. Herein, we present an overview of our recent understanding of the pathogenesis of bacterial IRGN, and emphasize the importance of NAPlr deposition and the related plasmin activity as general diagnostic biomarkers of bacterial IRGN. The concept of the present review is summarized in Figure 1. In general, clinical evidence in this field is very scarce, partly due to the nonexistence of internationally established diagnostic criteria and to the rarity of the disease. Furthermore, a reliable animal model completely mimicking human IRGN has not been established to date. Therefore, a substantial portion of this review is based on detailed evaluations of case reports.

## 2. Isolation and Characterization of NAPlr

We postulated that the nephritogenic protein for PSAGN should have affinity for a component in the serum of convalescent PSAGN patients [7]. Therefore, the fraction from the cytoplasmic proteins of GAS that demonstrates high affinity for the IgG of PSAGN patients was collected through affinity chromatography using PSAGN patients’ IgG-immobilized Sepharose. The fraction was then purified by ion exchange chromatography. Finally, a 43-kDa protein, identified to be a potent nephritogenic protein involved in PSAGN, was isolated [1,2].

The amino acid and nucleotide sequences of the isolated protein share high identity with those from the reported Plr [1,2] and glyceraldehyde-3-phosphate dehydrogenase (GAPDH) of GAS [8,9] as shown in Figure 2. Therefore, we named this protein NAPlr [1]. Streptococcal Plr is shown to bind plasmin in vitro, thus protecting it from physiologic inhibitors such as α2-antiplasmin and allowing it to maintain proteolytic activity [10]. NAPlr exhibited similar functions to streptococcal Plr, including specific binding of plasmin [11], GAPDH activity [9], and participating in the prominent alternative pathway (A-P) of complement activation as described in Step 5B below.

## 3. Clinical and Histological Characteristics of PSAGN in Relation to Complements, ICs, and NAPlr

PSAGN develops after streptococcal infection, with a latent period of approximately ten days. It is accompanied by decreased serum complement levels and glomerular deposition of C3 and/or IgG. Based on these characteristic manifestations, it is widely accepted that the immunological reaction against streptococcus related-antigens is involved in the etiopathogenesis of this disease [5,12,13,14].

As mentioned in the introduction, PSAGN is presumed to be an IC-mediated glomerulonephritis; however, glomerular immunoglobulin deposits are not always accompanied with C3 deposits, as shown in Figure 3A–D and Table 1 [2,3,4,15,16,17,18,19,20,21]. Furthermore, the site for the glomerular localization of ICs (subepithelial sites) and major glomerular inflammation (endocapillary) are completely different [2,6,11,12] as shown in Figure 4A–C and Figure 5A,B.

Direct immunofluorescence (IF) staining with rabbit anti-NAPlr antibody in renal biopsy tissue from PSAGN patients revealed glomerular NAPlr deposition mainly on the mesangium, the endothelial side of the glomerular basement membrane (GBM), and on neutrophils, appearing as a ring-like granular pattern as shown in Figure 6 and Figure 7A–C [1,2,6,7,11,12]. As shown in Table 2 [12], glomerular NAPlr deposition was observed in 100% (25/25) of cases in the early phase (1–14 days after onset) and in 61% (11/18) of cases in the proliferating phase (15–30 days after onset). However, no cases were positive for NAPlr in the resolving phase (31 days after onset). This indicates that the rate of positive NAPlr deposition decreased over time. In addition, the normal kidney was not positive for NAPlr, and only 4 out of 100 patients with other GNs were positive for NAPlr.

We suggest two possibilities regarding the mechanism by which NAPlr localizes on neutrophils. First, NAPlr could bind the urokinase-type plasminogen activator receptor expressed on neutrophils [25], which has been shown to be the receptor for streptococcal GAPDH [26]. Alternatively, NAPlr could be phagocytosed by neutrophils as a foreign bacterial antigen.

Another type of human GN with subepithelial IC deposition, namely membranous nephropathy, is rarely accompanied by endocapillary cell proliferation. Considering all these points, we explored the actual mechanism by which prominent glomerular endocapillary proliferation occurs and propose a pathogenic mechanism for the development of PSAGN.

## 4. Proposed Mechanism for PSAGN

Based on our findings on the nephritogenic protein responsible for PSAGN, we propose the following mechanisms for the development of the disease.


*Step 1*
*: Invasion of Streptococci into Host Tissues*


The ability to degrade tissue barriers formed by the extracellular matrix (ECM) and basement membrane (BM) is one of the most important factors in the pathogenesis of bacterial infection. The degradation of this network by secreted bacterial proteases leads to tissue and structural damage, which enhances the bacterial invasion in the host body. Figure 8 shows the direct binding pathway dependent on streptokinase and plasminogen [27], which is applicable to the invasion of GAS into respiratory tissues or skin. GAS can express high-affinity surface-binding molecules capable of capturing plasmin. Once bound, plasmin can no longer be regulated by the host regulators, particularly α2-antiplasmin [10]. This suggested a pathway by which GAS could both activate human plasminogen and capture the resulting enzymatically active product on their surface. A number of pathogenic bacteria produce plasminogen activators, and plasmin can also bind to surface receptors. Thus, the cell-surface localization of plasmin may be a common mechanism used by bacteria to facilitate movement through normal tissue barriers [28].


*Step 2: Deposition of NAPlr in the Glomeruli*


After a streptococcal infection of the respiratory tissues or skin, the nephritogenic protein NAPlr is released into circulation. Circulating NAPlr, likely in a free or fragmented form, accumulates on the inner side of the GBM, mesangial matrix, and infiltrating neutrophils as shown in Figure 9 (Step 2). NAPlr has been found to have high affinity for the ECM and multiple binding capacities for intracellular proteins such as lysozyme and cytoskeletal proteins. Indeed, NAPlr has a tendency to be deposited on the GBM, in the mesangial matrix, and in the cytoplasm of neutrophils [29].


*Step 3: Plasmin Trapped by Deposited NAPlr*


We suggest the following process by which plasmin can be trapped by deposited NAPlr. In the fluid phase (plasma), the host plasminogen activators, urokinase-type plasminogen activator and tissue-type plasminogen activator, and/or a streptococcal plasminogen activator such as streptokinase (SK), convert plasminogen into the active serine protease plasmin. Plasmin activity is strictly and rapidly controlled by its major physiological inhibitor, α2-antiplasmin. However, some plasmin would be trapped by the deposited NAPlr as shown in Figure 9 (Step 3). In the solid phase, NAPlr on the GBM would be immobilized, thus capturing plasmin and preserving its proteolytic activity. Whether or not plasmin could be trapped by deposited NAPlr would depend on ligand and receptor interactions and other participating factors that are not yet fully understood.


*Step 4: Plasmin-Related Endocapillary Glomerular Inflammation*


In contrast to the different distributions of NAPlr and C3/IgG, the glomerular distributions of plasmin activity and NAPlr are essentially identical, as shown in Figure 10, suggesting that deposited NAPlr could induce glomerular damage by trapping plasmin and preserving its activity [11]. Plasmin is involved in many physiological phenomena, including fibrinolysis, ECM turnover, cell migration, wound healing, angiogenesis, and neoplasia [31,32]. Moreover, plasmin activity can directly induce glomerular damage via the degradation of ECM proteins such as fibronectin and laminin, and could also exert an indirect effect on a variety of ECM proteins by activating pro-matrix metalloproteases [28]. In addition to this damage, plasmin mediates glomerular inflammation by triggering the accumulation and activation of monocytes, macrophages, and neutrophils, leading to plasmin-related endocapillary glomerular inflammation, as shown in Figure 9 (Step 4) [33,34,35].

We further investigated plasmin-related glomerular inflammation from the standpoint of cell-mediated immunity (CMI) [36,37] (Figure 11). First, in the initiation stage, deposited NAPlr at the glomerular endocapillary site directly activates the complement A-P, which releases complement-related anaphylatoxins, C3a and C5a, resulting in the glomerular endocapillary accumulation of macrophages and neutrophils. Plasmin also causes macrophages to accumulate in the same area [35]. Subsequently, NAPlr is recognized by immune cells, and CMI or a delayed type hypersensitivity (DTH) reaction against deposited NAPlr begin to take place, wherein T-helper/inducer cells (Th) interact with macrophages [38]. Second, activated macrophages release various growth factors, which promote endothelial and mesangial cell proliferation [39]. Finally, cell-to-cell contact occurs that further promotes the proliferation and amplification of in situ glomerular inflammation.


*Step 5A: IC-Dominant Glomerular Injury*


Circulating antibodies form IC, either in situ or in circulation, that can easily pass through the altered GBM and accumulate in the subepithelial space, appearing as “humps”. These final steps of immune cell accumulation and IC deposition cause the initial in situ glomerular inflammation to progress into a full-blown and overt disease state, as shown in Figure 12A,B (Step 5A). We measured circulating anti-NAPlr antibody levels using Western blot analysis (Table 3) [2]. Anti-NAPlr antibodies were more frequently detected in the sera of patients with PSAGN than in that of other subjects. Significantly higher levels of anti-NAPlr antibodies were found in the sera of PSAGN patients.

Subepithelial immune deposits form a “hump”, which is an accumulation of the complement components, ICs, and plasma proteins. In circulating and in situ formed ICs, NAPlr is fully saturated with the antibodies and complements (Figure 12B). Therefore, NAPlr may not be detected in the “humps”, either by IF microscopy or immuno-electron microscopy [40]. Andres et al. [41,42] also did not find antigen in the “humps”. However, Batsford et al. [43] reported that another potential nephritogenic protein, streptococcal pyrogenic exotoxin B (SPEB), was present in the “humps”. Antigen presence in these “humps” is a significant finding that will require additional investigation at the clinical stage (antigen or antibody excess) and the pathophysiological state (IC-dominant or C3-dominant).


*Step 5B: C3-Dominant Glomerular Injury*
(1)Complement profile in PSAGN


The activation of the complement system is a pivotal step in the pathogenesis of PSAGN. The activation of the classical pathway (C-P), the A-P, and the lectin pathway of the complement may be involved, depending on the clinical stage or pathophysiological state. A reduction in the circulating level of C3 is an almost universal feature of the acute phase of the disease, and levels usually return to normal within a month. As C1q and C4 are usually within normal limits, the reduction in C3 is assumed to be the result of the transient activation of the A-P of the complement system.

IF studies in patients with PSAGN have revealed that all PSAGN biopsy specimens obtained within 30 days of onset show intense and extensive deposition of C3 along the GBM and/or in the mesangium (Figure 3 and Figure 4). Glomerular C3 deposits with IgG were found in 27/43 cases (63%) and C3 without IgG in 16/43 cases (37%), 1–30 days after onset (Table 1). Oshima et al. [44] detected glomerular C3 deposits with IgG in 11/22 cases (50%) and C3 without IgG in 11/22 cases (50%), 1–36 days after onset. Similarly, other researchers have noted that “C3 deposits without IgG” exist in a substantial proportion of PSAGN patients. Subepithelial hump deposits were, however, equally observed in PSAGN patients with or without IgG deposits.
(2)Complement activation by NAPlr

The ability of NAPlr to activate the complement pathway has been previously studied [2]. C3 conversion into C3b was observed in human serum in vitro both in the presence or absence of chelating reagent (Figure 13A). Therefore, NAPlr can activate the A-P cascade in circulation. In addition, we found that NAPlr induced the formation of iC3b in a dose-dependent manner (Figure 13B).

Furthermore, we tried to establish an experimental model of PSAGN induced in rabbits using a similar potential nephritogenic protein called preabsorbing antigen (PA-Ag) [45]. PA-Ag is also a cytoplasmic protein previously purified from ruptured nephritogenic *Streptococci*, which we believe to be the same molecule as NAPlr, given that the molecular weight (43 kDa) and pI (4.7) of PA-Ag were the same as those of NAPlr. However, the complete full-length nucleotide sequence of PA-Ag has not yet been determined [46]. The experimental rabbits revealed slight to moderate endocapillary proliferative GN with exudative change, and showed diffuse glomerular staining for C3 without notable staining for IgG, as shown in Figure 14A,B. The evaluation of serial serum samples revealed a transient decrease in CH50 and C3 in the early phase. These experimental models were clinically and immunohistologically compatible with human PSAGN. These in vitro and in vivo studies thus suggest that NAPlr may act as an initiating factor and directly activate the A-P of the complement pathway. Moreover, the complement pathway can also be activated through ICs-induced C-P activation (Figure 15).
(3)C3-dominant glomerular injury

A C3-dominant pattern, that is, C3 deposits without the presence of IgG deposits, in PSAGN has been reported. Fish et al. [15] presented various explanations as to why C3 was present in the absence of IgG. They suggested that IgG could be obscured by C3, thus precluding its detection. Another explanation could be that C3 reached a detectable level for the IF technique, whereas IgG remained below the threshold level of detection. Regarding the absence of IgG, Satoskar et al. [47] suggested that “PSAGN may represent a transient form of C3 GN induced by streptococcal infection/antigens. In C3 GN, the glomerular deposits may not represent immunoglobulin containing ICs”. We agreed with their explanation and have proposed a mechanism for the overactivation of the A-P of the complement-mediated glomerular injury (GI) in PSAGN.

In 2013, Pickering et al. published a consensus report summarizing their findings on C3 glomerulopathy (C3G) [48] wherein a pathological entity characterized by C3 accumulation with absent, or sparse immunoglobulin depositions were introduced (Figure 16). Before and after this report, a variety of studies working on the same topic were published [49,50,51,52,53,54,55,56,57,58]. Cook et al. [59] postulated that “the deposition of C3 without immunoglobulin and the complement components associated with C-P activation implies an uncontrolled activation of the complement A-P. C3 GN, thus defined, is a distinct form of thrombotic microangiopathy, because in those cases complement activation is on the renal endothelium and is not associated with well-defined deposits on electron microscopy”. The described pathogenic mechanisms responsible for the development of C3G, including dense deposit disease (DDD) and C3 GN, are essentially the same as our proposed mechanisms for C3-dominant GI in PSAGN. Thus, we suspect the engagement of two distinct pathogenic mechanisms in the development of PSAGN: the previously mentioned IC-dominant GI and the C3-dominant GI.

The overlapping clinico-pathological features of C3G and post-infectious glomerulonephritis (PIGN) have been attracting considerable attention, and therefore, Al-Ghaithi et al. [22] evaluated 33 children with PIGN who underwent renal biopsies due to their unusual course. After pathological reclassification, 25 children were diagnosed with PIGN, and the remaining 8 (24.2%) were diagnosed with C3G. Cases of PIGN were further divided into Group A—C3 deposits with IgG (n = 16)—and Group B—C3 deposits without IgG (n = 9)—on routine IF. As expected, the outcome of patients with C3G was significantly worse than that of patients with PIGN. In contrast, they found no significant difference between Group A and Group B in terms of the clinical presentations and outcomes. In other words, the clinical features of PIGN patients with IgG deposition and without IgG deposition were essentially similar.

Both the IC-dominant and C3-dominant GI may alternately develop during in situ glomerular inflammation; however, we suspect that the C3-dominant GI would be more essential and universal than the IC-dominant GI. This is based on the histological evidence for the different localization of IC deposition and glomerular inflammation. Table 4 compares the characteristics of the IC-dominant GI and C3-dominant GI. The only major difference was the presence or absence of IgG under IF staining. We speculated on the possible reason for the development of the C3-dominant GI and why IgG would not be present. Initially, there must be an overactivation of the complement A-P by NAPlr with or without some other unknown initiating factors in the setting of the complement dysregulation, as shown in Figure 17A (Step 5B). This causes the amplification loop to turn over and over in the fluid phase, resulting in an overproduction of C3 breakdown products (i.e., iC3b, C3b, C3c and C3dg), which are assembled under the GBM. Subsequently, the breakdown products pass through the altered GBM and accumulate in the subepithelial space as “humps”. This process resembles IC-mediated GI, although the composition and structure of the humps varies. Here, IgGs forming NAPlr-IgG-ICs, IgG-complement components, and NAPlr-IgG-complement-ICs may become saturated with C3 breakdown products. C3 and C3b are known to be bound to the F(ab’)_2_ portion of IgG (Figure 17B) [60,61,62]. Therefore, under IF, C3c and C3dg should be positively stained, while IgG may not be stained. However, we speculate that IgG could be substantially present in the “humps” but saturated with complement components, resulting in negative staining. As seen in Table 4, this hypothesis is supported by previous findings showing that circulating ICs are highly increased in all patients with PSAGN [63]. In 2015, Messias et al. [64] performed studies to look for masked immunoglobulin deposits in 61 cases of immune complex-type diseases by the paraffin immunofluorescence after pronase digestion. Among them, 20 cases with negative to slight (0~1+) glomerular IgG staining by routine immunofluorescence turned to be positive to enhanced (2~3+) staining in the 18 cases. This study further supports our speculation. In addition to the above speculation, we would make it clear that IC deposition may similarly be formed in both GI groups, and that A-P complement activation may also be induced in both groups. However, IC-dominant GI may develop if the enhanced A-P is mild, while C3-dominant GI may occur if the enhanced A-P is strong.

The factors that overactivated the A-P in C3-dominant GI remain in question. Although they are not yet fully understood, we present several possibilities based on our previous findings: (1) As initiating factors, NAPlr may be released in circulation with other streptococcal components such as SPEB or streptococcal cell walls. SPEB is another potential nephritogenic protein previously described [43], and it has been known to directly activate the properdin-independent initiation of the complement A-P activation [65]; (2) As Satosker et al. suggested [47], “mild complement dysregulation may be involved”. It would be theoretically possible that “at least a subpopulation of patients with PSAGN have a mild genetic or acquired deficiency in one of the A-P complement regulatory proteins” that could trigger uncontrolled complement activation [66]; and (3) Autoantibodies to complement factors could have overactivated the A-P. As reported by Chauvet et al. [67], anti-factor B autoantibodies developed in PSAGN patients, which works to stabilize the amplification loop and maintain A-P activation. They were transiently found in up to 91% of PSAGN cases but only in 14% of C3 nephropathy cases. Autoantibodies to C3 convertase, C3NeF, were also reported to be transiently positive in 7/11 cases of atypical PSAGN [68].
ijms-23-09974-t004_Table 4Table 4Characteristic features of immune complex (IC)-dominant glomerular injury (GI) and C3-dominant GI.
IC-Dominant GIC3-Dominant GI
 **Major difference **
**C3 with IgG****C3 without IgG**

 **NAPlr deposition **
**+~++****+~++**

 **Plasmin activation **
**+~++****+~++**

 **Complement activation**




  **Alternative pathway**
**+****++**

  **Classical pathway**
**+****+**

 **Serum complement**



  CH50↓↓
  C3↓↓
  C4→→

 **Underlying complement dysregulation**
**unknown****unknown**

 **C3 Nephritic factor**
**+ ?****+ ?**[68]
 **Anti-factor B antibody**
**++ ?****++ ?**[67]
 **Pathology**




  **LM**



   Endocapillary GN**++****++**
   Exudative change**++****++**

  **IF**
   C3**++~+++****++~+++**
   IgG**+~++**−
   C4**+~****−**−

  **EM**
   Hump**++****++**

 **Circulating IC**
**++****++**[63]
 **Circulating NAPlr antibody**
**++****++**[2]C3 Nephritic factor and Anti-factor B antibody were reported by [67,68], respectively; however, PSAGN (post-streptococcal acute glomerulonephritis) was not classified as IC-mediated GI and C3-dominant GI, so “?” were added. This table summarizes the PSAGN (post-streptococcal acute glomerulonephritis) studies [1,2,6,11,12,24,36,37,40,44,45,46,63,69]. IC: immune complex, GI: glomerular injury, LM: light microscopy, IF: immunofluorescence, EM: electron microscopy, NAPlr: nephritis-associated plasmin receptor. Circulating IC: analyzed for C1q-binding activity, Circulating anti-NAPlr antibody: indicated in Table 3.

We have yet to come to a conclusion regarding what factors overactivated the A-P in C3-dominant GI, as there have not been enough published data or reports. Future research on PSAGN and C3 G particularly in those areas is necessary.

## 5. NAPlr as a Key Diagnostic Biomarker of IRGN

A typical case of PSAGN can be diagnosed relatively easily by confirming the characteristic findings, which include the onset of the disease in acute nephritic syndrome, a clinical course with a distinct latent period, abnormal urine findings with hematuria, C3-dominant hypocomplementemia, and elevated anti-streptolysin O (ASO). On the other hand, a recently emerging term “IRGN” has been proposed when infection is persistent at the time of GN onset in adult patients, especially in older patients. There are no internationally established diagnostic criteria for IRGN, including PSAGN. Nasr et al. [68] proposed the tentative diagnostic criteria for IRGN. In their proposed criteria, at least three of the following five items are required for the diagnosis of IRGN: (1) clinical or laboratory evidence of infection preceding or at the onset of GN, (2) depressed serum complement, (3) endocapillary proliferative and exudative GN, (4) C3-dominant or codominant glomerular IF staining, and (5) “hump”-shaped subepithelial deposits on electron microscopy. However, following these criteria, C3 glomerulopathy, membranoproliferative glomerulonephritis (MPGN), and lupus nephritis with endocapillary proliferative lesions would meet the requirement for the diagnosis, regardless of the presence or absence of infection. Thus, the proposed criteria described above is not perfect. As a matter of fact, there are no strict criteria in renal pathology to make the diagnosis. It is indeed sometimes difficult to make the differential diagnosis between C3GN and IRGN. Practically, however, only the combination of clinical history, laboratory data, and morphologic findings allows making the correct diagnosis. Although proof of prior infection is clinically important in diagnosing IRGN, it can be difficult to identify the causative organism, especially in the elderly with subclinical symptoms, unless a local bacterial culture is taken, the bacterial cultures are positive, or blood tests show an evident inflammatory reaction. Careful and thorough evaluation is required in the case of latent or deep-seated infections such as infective endocarditis or deep abscesses.

### 5.1. Glomerular NAPlr Deposition and Related Plasmin Activity in Streptococcal Infection-Related Nephritis

NAPlr was originally thought to be a useful marker only for the diagnosis of PSAGN, since its deposition in glomeruli is highly positive in patients with early PSAGN, as described in the previous sections. However, many cases of glomerular NAPlr-positive staining and associated plasmin activity in patients with various glomerular diseases other than PSAGN have been reported [30]. These cases include C3G [70,71] (Figure 18), MPGN type I [1,72,73], antineutrophil cytoplasmic antibody (ANCA)-associated vasculitis, both ANCA-positive [74] and negative [75], and IgA vasculitis [76]. The glomerular staining patterns of NAPlr and plasmin activity in these cases were quite similar, which made it difficult to differentiate them from PSAGN. Furthermore, most of these cases have common features, such as elevation in serum ASO titers, histological findings of endocapillary proliferation on light microscopy, and subepithelial electron-dense deposits on electron microscopy. Based on these findings, we suspected the existence of group of glomerular diseases in which streptococcal infection has led to the glomerular deposition of NAPlr and subsequent upregulation in related plasmin activity, leading to the development of endocapillary proliferative glomerular lesions. We termed such a condition as streptococcal infection-related nephritis (SIRN), which includes PSAGN [12]. The clinical and pathological profiles of patients with SIRN are summarized in Table 5. As will be discussed later, the concept of SIRN has now expanded, and the term ‘IRGN’ has become widely known following the report of Nasr et al. [69]. If the diagnostic criteria proposed by Nasr et al. are applied, all of the cases with SIRN would be diagnosed as IRGN.

### 5.2. Positive Staining for NAPlr/Plasmin Activity in Non-Streptococcal IRGN

Even when we proposed the disease entity of SIRN, we still believed that NAPlr was specific for *Streptococcus*
*pyogenes*. However, as we were analyzing patients with IRGN, we discovered some IRGN patients with positive NAPlr/plasmin activity staining that were induced by the infections other than *Streptococcus*
*pyogenes*, such as *Streptococcus pneumoniae* [77], *Aggregatibacter actinomycetemcomitans* [78], *Mycoplasma pneumoniae* [79], and *Staphylococcus aureus* (both methicillin-sensitive and resistant strains; unpublished case in preparation) (Table 6). NAPlr is the same substance as the GAPDH of *Streptococcus* as described in Section 2 of this manuscript, and GAPDH is universally expressed and may have high homology among various bacteria. Therefore, anti-NAPlr antibodies are likely to cross-react with the GAPDH of bacteria other than *Streptococcus*
*pyogenes*. It is also possible that GAPDH from other species have plasmin-binding capacity. As shown in Table 7, *Streptococcus pneumoniae* GAPDH shares high identity with NAPlr in terms of the amino acid sequence, and the C-terminal sequences, which are the most likely to be associated with the plasmin-binding capacity, are completely identical to those of streptococcal GAPDH or NAPlr [77]. The sequences of GAPDH from *A. actinomycetemcomitans* and *S. aureus* also show high similarity to the sequence of NAPlr; moreover, the latter also have a C-terminal lysine. *M. pneumoniae* GAPDH has been shown not only to have cross-immunoreactivity to the anti-NAPlr antibody but also to have a plasmin-binding function (case 3 in Table 6) (Figure 19) [79]. In addition, *M. pneumoniae* GAPDH has been reported to bind to plasminogen and convert it to plasmin [80].

### 5.3. Plasmin-Binding Capacity of GAPDH in Non-Streptococcal Bacteria

The C-terminal lysine residue of the plasmin receptor appears to be essential for the plasmin-binding activity [81]. However, in 1998, Winram et al. [82] analyzed this fact using site-directed mutagenesis of streptococcal plasmin-receptor protein. They substituted Lys334 residue with a leucine codon and found that the replacement did not reduce plasmin-binding capacity. This indicated that the plasmin receptor did not function as a sole plasmin-binding receptor, but other molecules were also responsible for the plasmin-binding phenotype of GAS. As mentioned above, the GAPDH of some bacteria that induced non-streptococcal IRGN had the C-terminal lysine residue as the plasmin-binding site. Both plasmin and plasminogen molecules have lysine-binding sites. In particular, they contain five homologous Kringle domains, which bind to C-terminal lysine on substrates.

### 5.4. NAPlr as a General Diagnostic Biomarker of IRGN

Positive staining for NAPlr and associated plasmin activity has been used as a marker of IRGN even when the causative organism or site of infection cannot be identified, possibly due to deep-seated infections or the use of antimicrobial agents. In fact, there have been reports of patients with various forms of GN even if the causative pathogens or the infection sites could not be identified (Table 8). These cases include proliferative glomerulonephritis with monoclonal immunoglobulin G deposits (PGNMID) [83] and eosinophilic proliferative glomerulonephritis [84], C3G [85], small vessel vasculitis [86], IgA nephropathy [87], and IgA-dominant IRGN [88]. Positive staining for NAPlr and plasmin activity in these cases offer the etiologic evidence for the contribution of prior or ongoing infection.

Takehara et al. [83] reported a case of PGNMID associated with infection and a genetic mutation in complement factor H, which presented with rapidly progressive GN requiring hemodialysis, nephrotic syndrome, and diffuse proliferative and crescentic GN (case 7 in Table 8). Antibiotic treatment was chosen due to the results of positive glomerular staining for NAPlr and plasmin activity (Figure 20), suggesting a preceding or ongoing infection in the first biopsy. Spontaneous recovery of renal function and urine protein were achieved without immunosuppressive therapy. A second biopsy, which was performed when the patient developed nephrotic-range proteinuria one year later, showed that NAPlr/plasmin activity staining were negative. Thus, glomerular staining for NAPlr and plasmin activity could help in diagnosis and prevent unnecessary immunosuppressive treatment. It is beneficial and essential to clarify the involvement of a preceding or ongoing infection by examination for NAPlr/plasmin activity in these cases. Noda et al. [89] reported a case of asymptomatic sinusitis as a cause of IRGN which clinically presented with hypocomplementemic nephrotic syndrome and kidney dysfunction, histopathologically showing MPGN type I (case 5 in Table 8). Immunosuppressive treatment with a combination of steroids and cyclosporin was initiated, but the clinical symptoms did not improve. Staining of kidney specimens revealed positive staining for NAPlr and plasmin activity in glomeruli (Figure 21). Thus, IRGN due to a hidden infection was suspected. Thorough and careful examinations, including a computed tomography (CT) scan, eventually revealed sinusitis in his left maxillary sinus, suggesting that asymptomatic sinusitis contributed to the onset of GN. With the proper antibiotic treatment, the nephrotic syndrome went into remission, and there was improved hypocomplementemia together with the opacification resolution of the sinuses. Thus, NAPlr-positive staining led to the diagnosis of the GN triggered by infection, and the treatment of the infected sites resulted in the remission of the clinical manifestations. Furthermore, positive staining for NAPlr and plasmin activity were also observed in the pulmonary artery in the report by Okabe et al. [84] (case 8 in Table 8) (Figure 22). Thus, NAPlr/plasmin activity staining may be useful for the assessment of not only the kidneys but also other organs.

As described above, NAPlr and related plasmin activity were positive not only in PSAGN, but also in cases of SIRN with other clinical and histopathological diagnoses, and in cases of IRGN caused by various infections. Thus, the positive results of this staining strongly suggest that some kind of infection was involved in the pathogenesis of the GN. In this regard, staining for NAPlr and plasmin activity is thought to be a general diagnostic marker of IRGN, which could support the diagnosis of IRGN from an etiological perspective.

## 6. Conclusions

In this review, we summarized our discoveries on the causative agent for PSAGN, NAPlr, and its principal character and function. NAPlr was isolated from cytoplasmic proteins of GAS, and was found to be the same molecule as the GAPDH of GAS. It has the functions of plasmin-binding capacity and direct activation of complement A-P. With these functions, NAPlr deposited in endocapillary sites induces plasmin-related and complement A-P-related endocapillary glomerulonephritis.

The pathogenic mechanism of PSAGN has been described mainly in relation to the humoral immune mechanism, IC-dominant GI. However, clinical and histological evaluation including the streptococcal nephritogenic protein, NAPlr, revealed a more essential contribution of C3-dominant GI to the development of PSAGN by the overactivation of the complement A-P. This C3-dominant GI closely resembles to C3 glomerulopathy. We proposed an essential engagement of complement A-P-mediated GI, and postulated the saturation theory for IgG as a result of overproduction of C3 breakdown products. We considered two possibilities to complete the theory. The first one is, as partly mentioned above, the pronase-digested paraffin immunofluorescence method, which Nasr et al. [90,91,92] and Messias et al. [64] advocated as the important technique to salvage the masked deposits. Larsen et al. [93,94], and others [95,96,97,98] have found the masked monotypic immunoglobulin deposits in various GN. The second possibility is laser dissection and mass spectrometry (LCMS) proteomic analysis. Sethi et al. [49,55,68,99] have performed LCMS to determine the proteomic profile in diverse diseases, in which they measured complement components, immunoglobulins, and other factors. Thus, we expect that glomerular bound IgG could be measured by LCMS and quantitatively compared the level of bound IgG in between IC-dominant GI and C3-dominant GI in PSAGN.

The biggest problem with research in this field would be the nonexistence of internationally established diagnostic criteria. Without the diagnostic criteria and with few patients, it is difficult to establish clear clinical evidence in this field. Furthermore, a reliable animal model completely mimicking human IRGN has not been established to date. Therefore, the evaluation of each case is quite important and essential to understanding the mechanisms of the IRGN disease process. The disease framework of IRGN has been further expanded. Herein, we introduced recently reported important cases with IRGN. The GAPDH of various bacteria other than *Streptococci* have been found to react with anti-NAPlr antibodies, thus suggesting cross-immunoreactivity with anti-NAPlr antibodies. We thus realized that positive staining for NAPlr and plasmin activity is strongly indicative of prior or ongoing infection. Controlling the underlying infection and managing developed GN are essential for the effective treatment of IRGN; however, the prompt diagnosis of IRGN is often difficult, as specific diagnostic biomarkers have not yet been identified. Through this review, we highlight that NAPlr deposition and the related plasmin activity in the glomeruli should be utilized as general diagnostic biomarkers of IRGN.

## Figures and Tables

**Figure 1 ijms-23-09974-f001:**
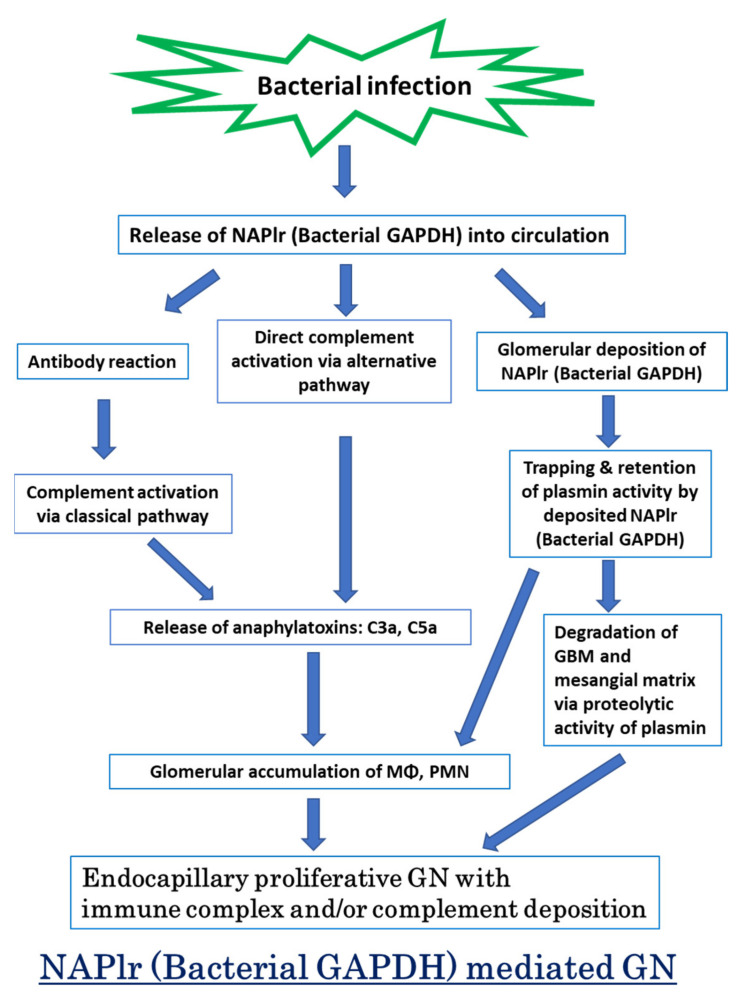
Graphical abstract of this review. NAPlr: nephritis-associated plasmin receptor, GAPDH: glyceraldehyde-3-phosphate dehydrogenase, MΦ: macrophage, PMN: polymorphonuclear neutrophil, GBM: glomerular basement membrane, GN: glomerulonephritis.

**Figure 2 ijms-23-09974-f002:**
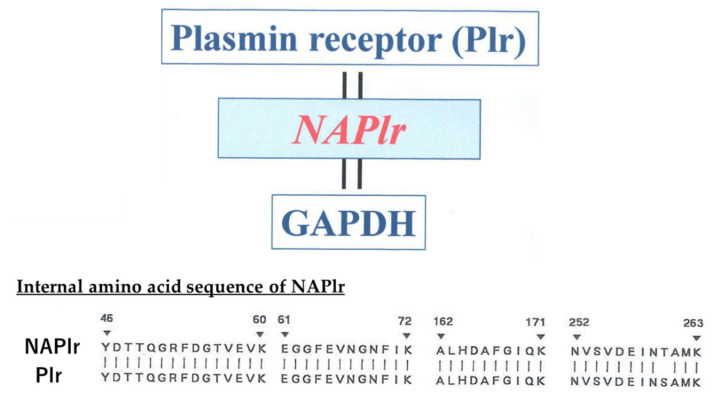
The amino acid and nucleotide sequences of NAPlr share high sequence identity with streptococcal Plr, as shown in the internal amino acid sequence of NAPlr. Because Plr and GAPDH are already established to be the same molecules, these three proteins are considered identical. NAPlr: nephritis-associated plasmin receptor, GAPDH: glyceraldehyde-3-phosphate dehydrogenase, Plr: plasmin receptor.

**Figure 3 ijms-23-09974-f003:**
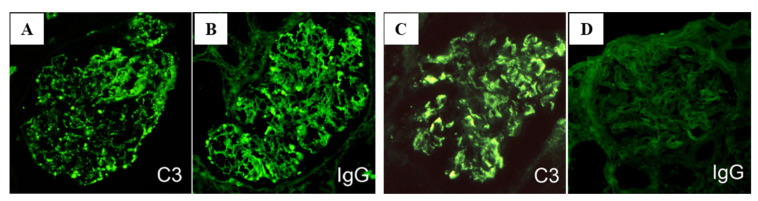
Two different patterns of glomerular C3 and IgG deposits in post-infectious glomerulonephritis. (**A**,**B**): C3 with IgG deposits, positive granular staining for C3 (**A**) and for IgG (**B**). (**C**,**D**): C3 without IgG deposits, positive granular staining for C3 (**C**) but completely negative staining for IgG (**D**). Original magnification: ×400. Photographs are cited from a previously published report [22].

**Figure 4 ijms-23-09974-f004:**
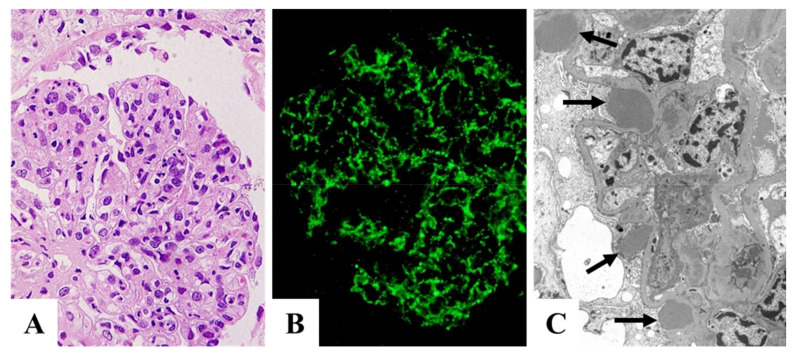
Characteristic pathology in the early phase of PSAGN. (**A**): Light microscopy (HE); endocapillary proliferative glomerulonephritis. Original magnification: ×400. (**B**): C3 staining; fine granular deposits mainly on capillary wall, part of mesangium. Original magnification: ×400. (**C**): Electron microscopy; characteristic subepithelial dome-shaped humps (indicated by black arrows). PSAGN: post-streptococcal acute glomerulonephritis. Photographs are cited from a previously published textbook [23], where details of all the staining methods and magnifications have been described.

**Figure 5 ijms-23-09974-f005:**
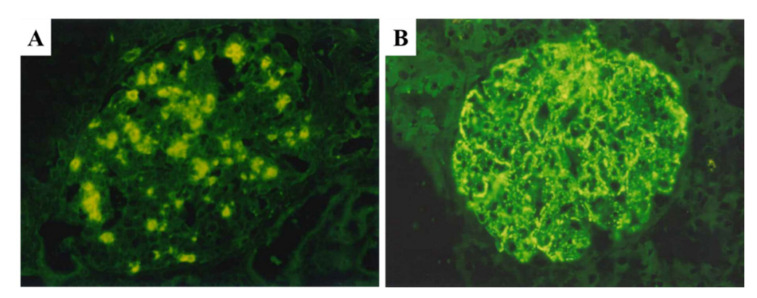
Difference in glomerular localization of NAPlr and immune complexes. (**A**): In the early phase, localization of NAPlr, which is presumed to be a free antigen, is primarily on the mesangium, part of the GBM, and infiltrating neutrophils. (**B**): In the early phase, C3 is localized mainly along the GBM (subepithelial), and partly on the mesangium. Original magnification: ×400. GBM: glomerular basement membrane, NAPlr: nephritis-associated plasmin receptor. Photographs are cited from a previously published report [2], where details of all the staining methods have been described.

**Figure 6 ijms-23-09974-f006:**
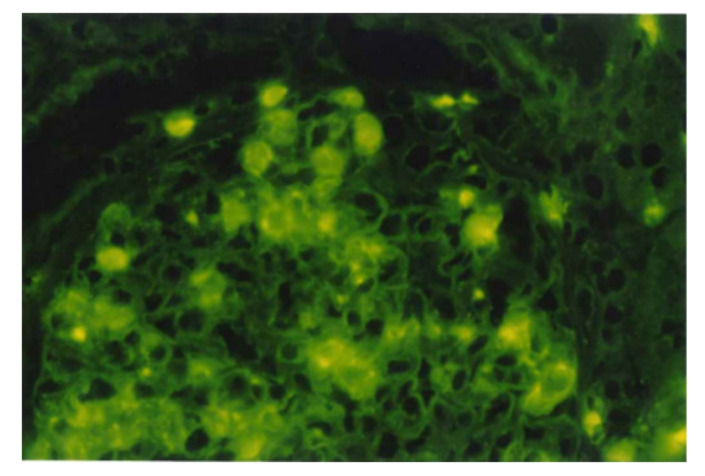
Immunofluorescence staining with rabbit anti-NAPlr antibody in glomeruli from a PSAGN patient (10 days after onset) revealed glomerular NAPlr deposition mainly on the mesangium, the endocapillary side of GBM, and infiltrating neutrophils in a ring-like granular pattern. Original magnification: ×400. GBM: glomerular basement membrane, NAPlr: nephritis-associated plasmin receptor, PSAGN: post-streptococcal acute glomerulonephritis. This photograph is cited from a previously published report [24].

**Figure 7 ijms-23-09974-f007:**
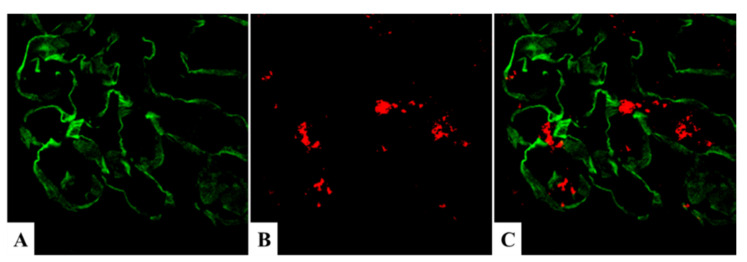
NAPlr localization relative to the glomerular basement membrane. Confocal microscopy images of double immunofluorescence staining for (**A**) α3 chain of collagen IV (FITC), (**B**) NAPlr (Alexa Fluor 594), and (**C**) the merged image in a post-streptococcal acute glomerulonephritis patient. NAPlr immunofluorescence is mostly seen on the inner side of glomerular tufts. Original magnification: ×400. NAPlr: nephritis-associated plasmin receptor. Photographs are cited from a previously published report [6], where details of all the staining methods have been described.

**Figure 8 ijms-23-09974-f008:**
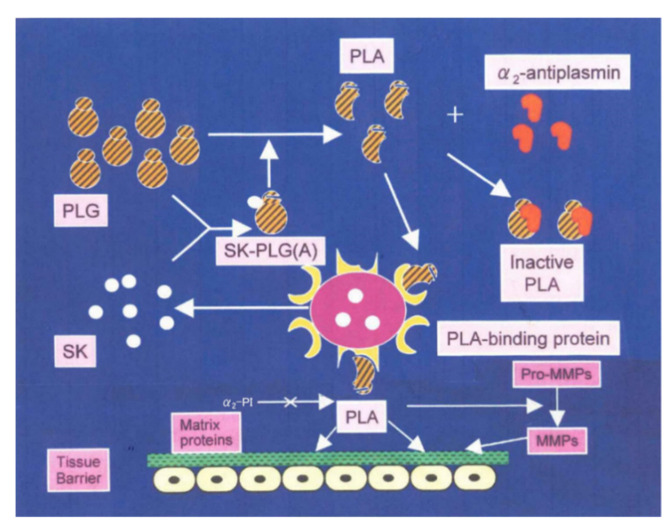
Reported mechanism for the tissue invasion of *Streptococcus* through direct binding pathway dependent on streptokinase and plasminogen. PLG: plasminogen, PLA: plasmin, SK: streptokinase, MMPs: matrix metalloproteases.

**Figure 9 ijms-23-09974-f009:**
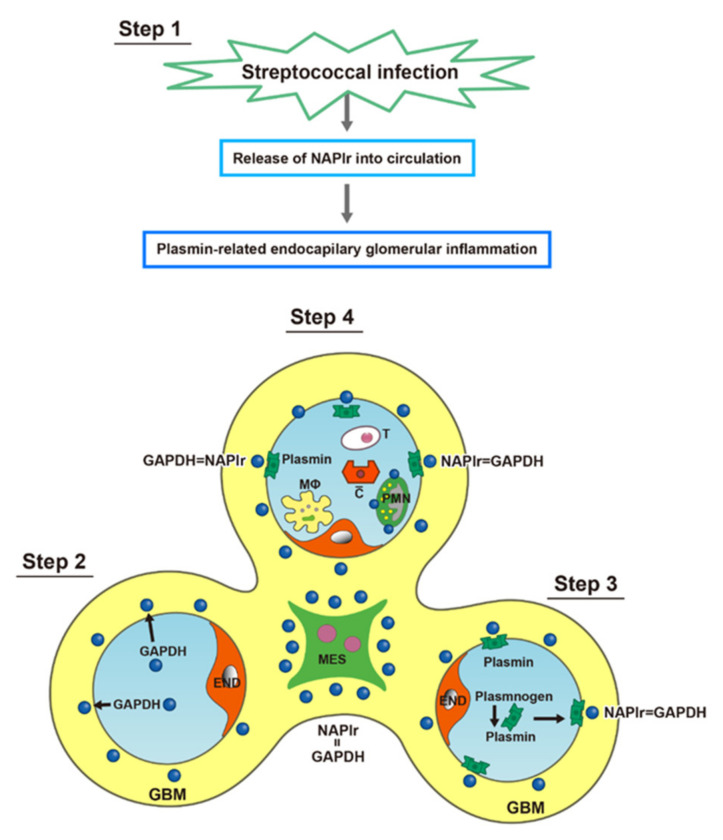
Schematic representation of the proposed mechanism for PSAGN. GAPDH: glyceraldehyde-3-phosphate dehydrogenase, GBM: glomerular basement membrane, END: endothelial cell, MES: mesangium, NAPlr: nephritis-associated plasmin receptor, MΦ: macrophage, PMN: polymorphonuclear neutrophil, PSAGN: post-streptococcal acute glomerulonephritis. This schema is modified and cited from a previously published paper [30].

**Figure 10 ijms-23-09974-f010:**
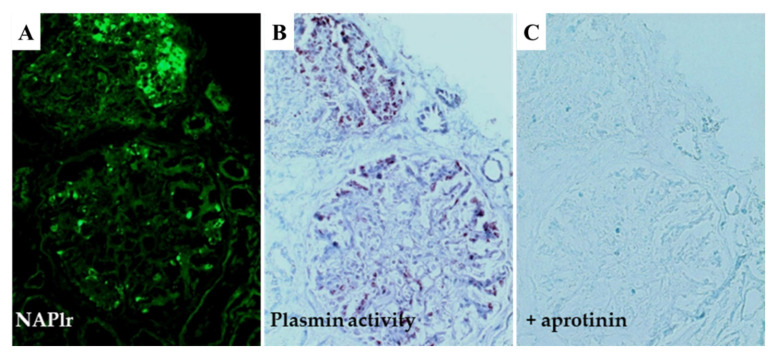
In situ zymography for plasmin activity. (**A**) Positive NAPlr staining in the glomeruli of a PSAGN patient 18 days after onset. (**B**) Prominent plasmin activity was observed in a similar distribution to NAPlr deposition in the same glomeruli as shown in (**A**). This activity was completely inhibited by the addition of aprotinin (**C**). NAPlr: nephritis-associated plasmin receptor, PSAGN: post-streptococcal acute glomerulonephritis. Photographs are cited from a previously published report [11], where details of all the staining methods have been described. Original magnifications: ×200.

**Figure 11 ijms-23-09974-f011:**
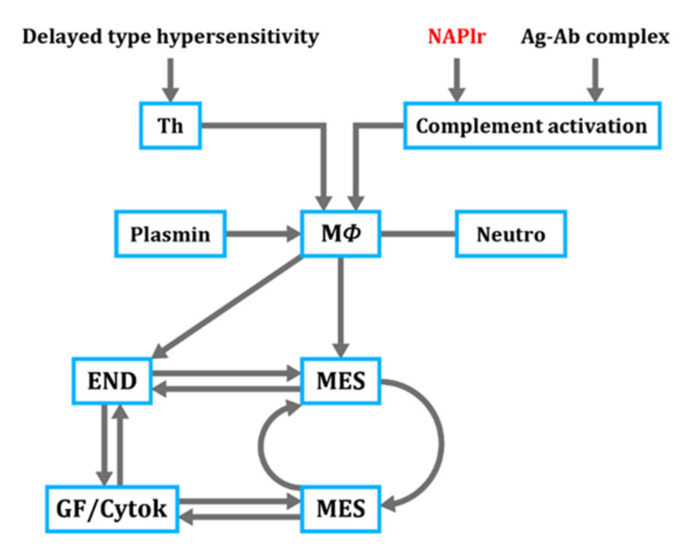
Cell-mediated immune response in PSAGN focused on NAPlr. Th: T helper/inducer cell, Ag-Ab complex: antigen-antibody complex, MΦ: macrophage, Neutro: neutrophil, END: endothelial cell. MES: mesangial cell, END: endothelial cell, GF/Cytok: growth factor/cytokines, NAPlr: nephritis-associated plasmin receptor, PSAGN: post-streptococcal acute glomerulonephritis.

**Figure 12 ijms-23-09974-f012:**
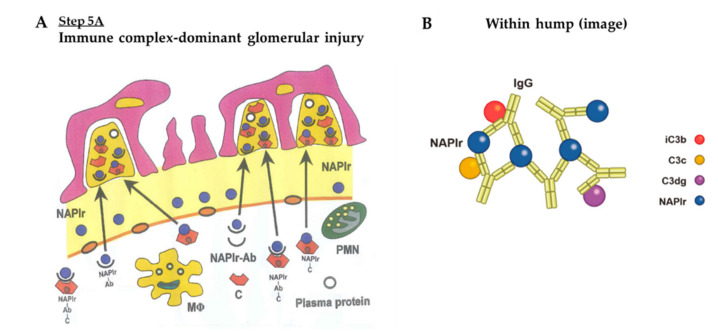
(**A**) Immune complex-dominant glomerular injury. (**B**) NAPlr is fully saturated with IgG and complement components, however, IgG is not saturated with complement components. NAPlr: nephritis-associated plasmin receptor, NAPlr-Ab: NAPlr-antibody complex, NAPlr-Ab-C: NAPlr-antibody-complement complexes, MΦ: macrophage, PMN: polymorphonuclear neutrophil, C: complement, iC3b: inactivated C3b, NAPlr: nephritis-associated plasmin receptor.

**Figure 13 ijms-23-09974-f013:**
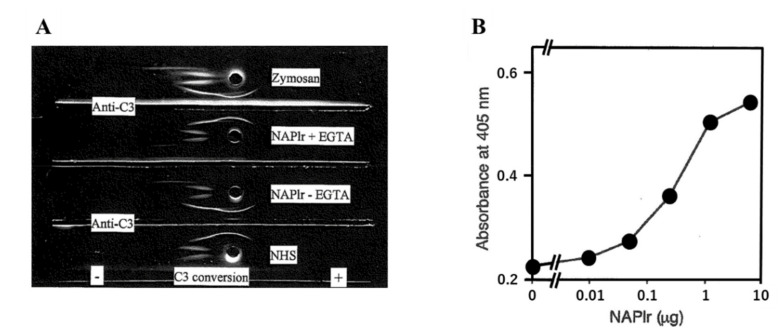
Complement activation by NAPlr. (**A**) Immunoelectrophoresis showing conversion of C3 after incubation of NHS with NAPlr with or without Mg^2^⁺ and EGTA (middle). As a possible control, zymosan was added to NHS and indicates activated C3 (top). NHS not incubated with NAPlr shows a single arc of C3 (bottom). (**B**) Formation of iC3b from C3 in NHS incubated with various amounts of NAPlr. The plots use average values from triplicate assays. NAPlr: nephritis-associated plasmin receptor, EGTA: ethylene glycol tetraacetic acid, NHS: normal human serum.

**Figure 14 ijms-23-09974-f014:**
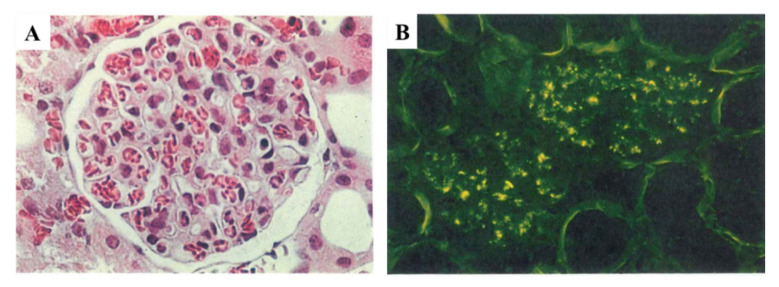
(**A**): Representative light microscopy image of experimental glomerulonephritis induced by the administration of streptococcal nephritogenic protein in rabbits, 4 weeks after the start of injection. Moderate endothelial and mesangial proliferative changes with prominent infiltrating leucocytes are evident (H&E, original magnification ×400). (**B**): Immunofluorescence of glomeruli from representative rabbits in experiment, 4 weeks after the start of injections. C3 is intensely deposited along the glomerular basement membrane and mesangial areas (FITC-labelled goat anti-rabbit C3, original magnification ×200). Photographs are cited from a previously published report [45], where details of all the staining methods have been described.

**Figure 15 ijms-23-09974-f015:**
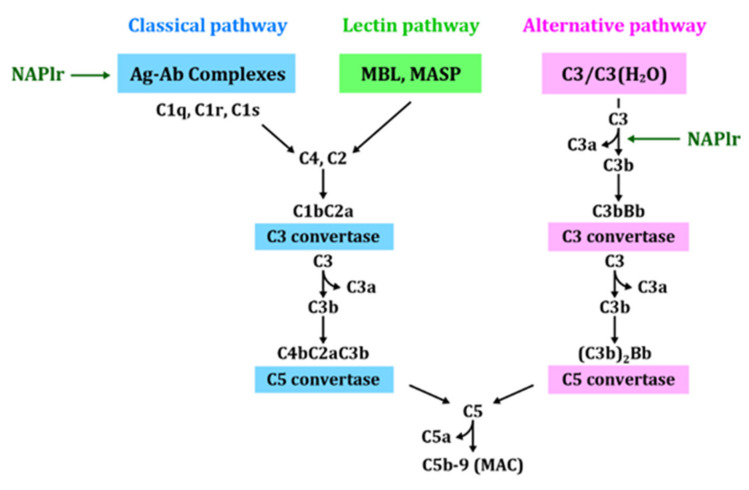
Schematic representation of the complement activation through each pathway and NAPlr. Ag-Ab: antigen-antibody, MBL: mannose-binding lectin, MASP: mannose-binding lectin-associated serine proteases, C3(H₂O): hydrolyzed C3, MAC: membrane attack complex, NAPlr: nephritis-associated plasmin receptor.

**Figure 16 ijms-23-09974-f016:**
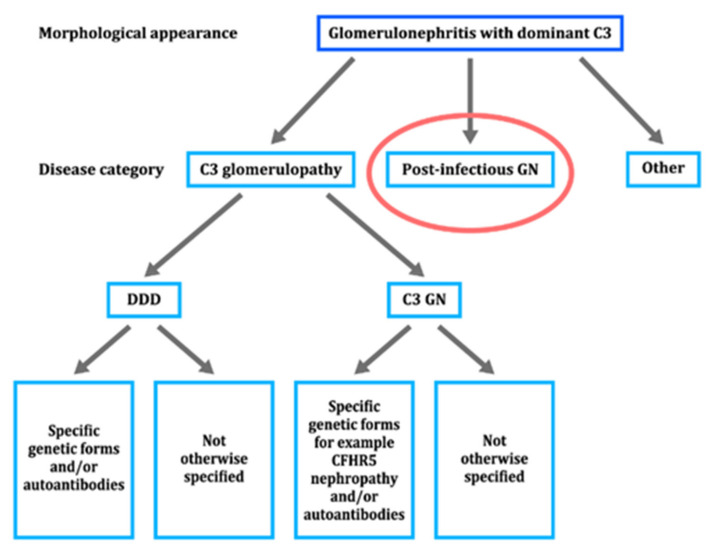
A schematic diagram showing an approach to the classification of disease in a biopsy specimen, showing the morphological changes of glomerulonephritis with dominant C3. Post-infectious glomerulonephritis (GN) highlighted by red circle should be differentiated from C3 glomerulopathy. DDD: dense deposit disease, C3 GN: C3 glomerulonephritis.

**Figure 17 ijms-23-09974-f017:**
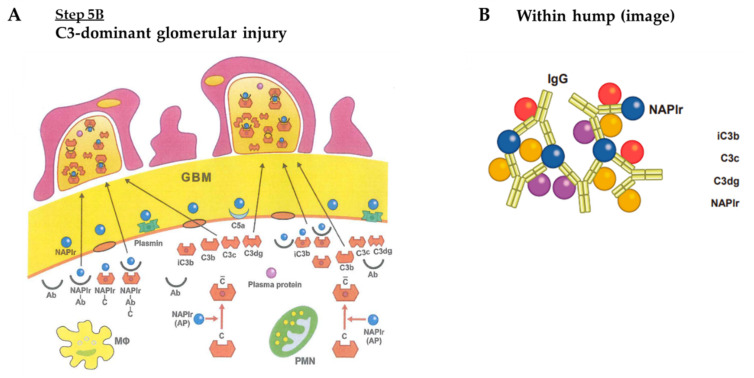
(**A**): C3-dominant glomerular injury. (**B**): NAPlr is fully saturated with IgG and complement components, and IgG is also fully saturated with complement components. Ab: antibody, NAPlr-Ab: NAPlr-antibody, NAPlr-C: NAPlr-complement complex, NAPlr-Ab-C: NAPlr-antibody-complement complexes, C bar: activated complement, iC3b: inactivated C3b, A-P: alternative pathway, MΦ: macrophage, PMN; polymorphonuclear neutrophil, NAPlr: nephritis-associated plasmin receptor.

**Figure 18 ijms-23-09974-f018:**
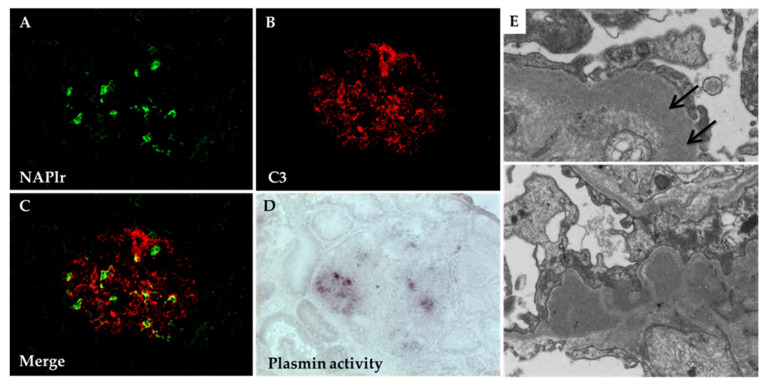
Kidney biopsy findings of C3 glomerulopathy with glomerular NAPlr deposition and associated plasmin activity. (**A**–**C**): Double immunofluorescent staining for NAPlr (fluorescein isothiocyanate, green) and complement C3 (Alexa Fluor 594, red). NAPlr (**A**) and C3 (**B**) were both positive in the glomeruli, and their localization was essentially different in the merged images (**C**). (**D**): By in situ zymography, the plasmin activity was found to have a distribution similar to that of the NAPlr staining in the glomeruli. Original magnifications of (**A**–**D**): ×200. (**E**): (Upper panel) Massive electron-dense deposits (arrow) were observed along the lamina densa in the glomerular basement membrane (GBM). (Lower panel) Subepithelial “hump”-shaped electron-dense deposits were also present in the GBM. Photographs are cited from a previously published report [71], where details of all the staining methods have been described.

**Figure 19 ijms-23-09974-f019:**
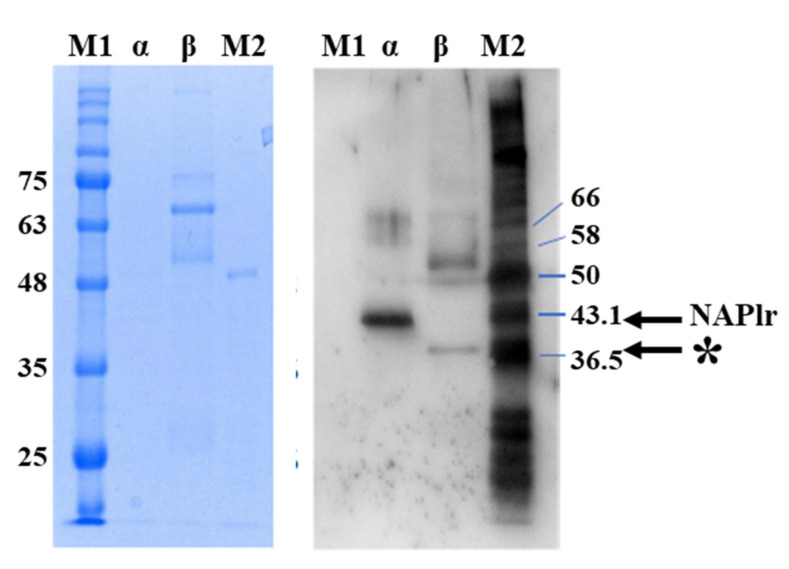
Western blotting analysis of isolated bacterial protein. Protein staining of the blot of isolated protein from *M. pneumoniae* and *Streptococcus pyogenes* (left) and Western blotting using anti-NAPlr antibody (right). Western blotting revealed a specific band in the β-lane at around the predicted size of *M. pneumoniae* GAPDH (37 kDa: indicated by * with arrow), confirming the cross-reactivity of the anti-NAPlr antibody with *M. pneumoniae* GAPDH. The strong band in the α-lane with a molecular mass of 43 kDa, indicated by the arrow, is the NAPlr protein. Lanes: M1, prestained marker; α = isolated protein of *S. pyogenes*; β = isolated protein of *M. pneumoniae*; M2, Western 7 Protein Ladder [79].

**Figure 20 ijms-23-09974-f020:**
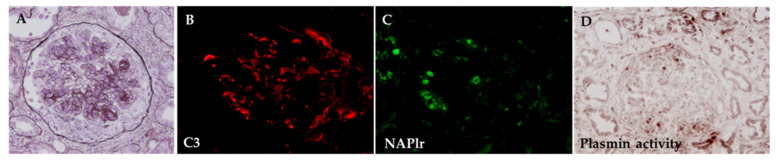
Kidney biopsy findings for case 7 in Table 8. (**A**): Mesangial and endocapillary proliferation with double contours along the capillary walls and a cellular crescent (periodic acid-methenamine-silver stain, original magnification ×200). C3 (**B**) and NAPlr (**C**) were both positive in glomeruli, and their localization was essentially different. (**D**): In situ zymography showed that the glomerular distribution of plasmin activity was similar to that of NAPlr staining. Photographs are cited from a previously published report [83], where details of all the staining methods have been described.

**Figure 21 ijms-23-09974-f021:**
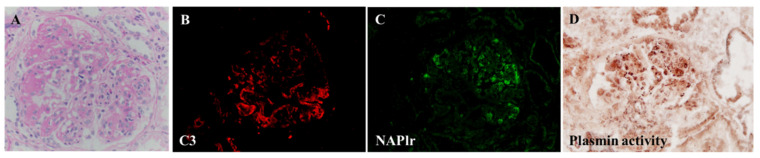
Kidney biopsy findings for case 5 in Table 8. (**A**) Light microscopy showing diffuse proliferative and exudative glomerulonephritis with infiltrating neutrophils (periodic acid–Schiff, original magnification ×400). (**B**,**C**) C3 (**B**) and NAPlr (**C**) were both positive in glomeruli, and their localization was essentially different (**B**,**C**). (**D**) In situ zymography showed that the plasmin activity was similar to the NAPlr staining in the glomeruli in its distribution. Photographs are cited from a previously published report [89], where details of all the staining methods have been described.

**Figure 22 ijms-23-09974-f022:**
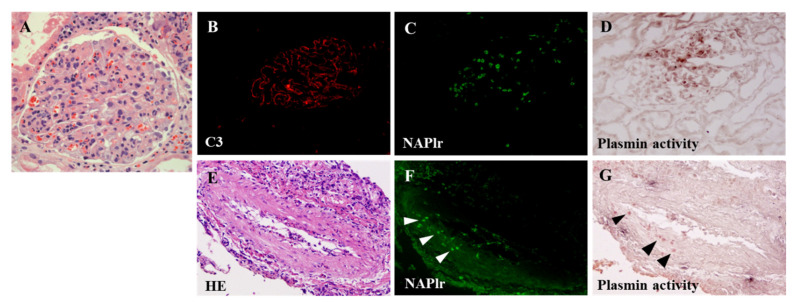
Kidney biopsy findings for case 8 in Table 8. (**A**): Light microscopy showing endocapillary proliferation with massive eosinophil infiltration (H&E, original magnification ×400). (**B**): Immunostaining for C3. C3 was stained granularly along the glomerular tuft. NAPlr was positively stained in the glomeruli (**C**), and the distribution of plasmin activity (**D**) was similar to that of NAPlr staining in the glomeruli. (**E**–**G**): Eosinophils were observed to infiltrate in the wall of the pulmonary artery, where NAPlr positive immunostaining and strong plasmin activity were detected (arrowheads) in addition to in the glomerulus. Photographs are cited from a previously published report [84], where details of all the staining methods have been described.

**Table 1 ijms-23-09974-t001:** Immunofluorescence studies in patients with PSAGN.

Time from Onset to Biopsy	n	NAPlr	Plasminogen	Fibrinogen	C3	IgG	IgA	IgM
1–14 d	25	25/25 (100%)	10/25 (40)	15/25 (60)	25/25 (100)	16/25 (64)	11/25 (44)	10/25 (40)
15–30 d	18	11/18 (61)	5/18 (28)	11/18 (61)	18/18 (100)	11/18 (61)	8/18 (44)	9/18 (50)
31–90 d	7	0/7 (0)	0/7 (0)	4/7 (57)	6/7 (86)	3/7 (43)	3/7 (43)	3/7 (43)
Total	50	36/50 (72)	15/50 (30)	30/50 (60)	49/50 (98)	30/50 (60)	22/50 (44)	22/50 (44)
**Time from Onset to** **Biopsy**	**n**	**P**	**C1q**	**C4**	**C5**	**C9**	**S**	**MAC**
1–14 d	25	23/25 (92%)	7/25 (28)	8/25 (32)	25/25 (100)	24/25 (96)	24/25 (96)	25/25 (100)
15–30 d	18	16/18 (89)	6/18 (33)	3/18 (17)	18/18 (100)	17/18 (94)	17/18 (94)	18/18 (100)
31–90 d	7	5/7 (71)	2/7 (29)	1/7 (14)	6/7 (86)	5/7 (71)	5/7 (71)	6/7 (86)
Total	50	44/50 (88)	15/50 (30)	12/50 (24)	49/50 (98)	46/50 (92)	46/50 (92)	49/50 (98)

NAPlr: nephritis-associated plasmin receptor, P: properdin, S: S protein, MAC: membrane attack complex, C3 represents C3c, Percentages are reported in parentheses.

**Table 2 ijms-23-09974-t002:** Glomerular NAPlr positivity in PSAGN, non-PSAGN, and normal kidneys.

Biopsy Specimens	Onset to Biopsy	Glomerular NAPlr (+)	
PSAGN	1–14 days	25/25	(100%)
	15–30 days	11/18	(61%)
	31–90 days	0/7	(0%)
	Total	36/50	(72%)
Non-PSAGN		4/100	(4%)
Normal kidneys		0/10	(0%)

NAPlr: nephritis-associated plasmin receptor, PSAGN: post-streptococcal acute glomerulonephritis.

**Table 3 ijms-23-09974-t003:** Anti-NAPlr antibody in the patients with PSAGN, Group A streptococcal infection without renal involvement, pediatric children, and normal adults [2].

	Age in Years Range	Anti-NAPlr Antibody (+)	Anti-NAPlr Antibody Titers
PSAGN	(5–72 y, mean 29.3)	46/50 (92%)	566 ± 106.1
Streptococcal infection	(8–64 y, mean 29.0)	30/50 (60%)	227.1 ± 51.2
Pediatric I	(0.2–10 y, mean 7.2)	13/50 (26%)	138.9 ± 23.4
Pediatric II	(11–20 y, mean 14.1)	18/50 (36%)	166.0 ± 25.7
Normal adults I	(25–35 y, mean 30.0)	24/50 (48%)	100.1 ± 18.0
Normal adults II	(52–59 y, mean 53.0)	36/50 (72%)	186.0 ± 17.3

NAPlr: nephritis-associated plasmin receptor, PSAGN: post-streptococcal acute glomerulonephritis. Pediatric I: group of younger children (27 boys and 23 girls), Pediatric II: group of older children (23 boys and 27 girls), Normal adults I: group of younger adults (25 men and 25 women), Normal adults II: group of older adults (25 men and 25 women). These groups matched with patients with PSAGN in terms of sex and age, and they showed no signs of recent streptococcal infection.

**Table 5 ijms-23-09974-t005:** Clinical and pathological profiles of patients with SIRN.

Case	Age/ Gender	Diagnosis	Onset	UP	U-RBC (/hpf)	SCr (mg/dL)	CH50 (U/mL)	C3 (mg/dL)	C4 (mg/dL)	CH50 (U/mL)	EP	IF	NAPlr	Ref.
1	14/F	DDD	AGN	+++	+++	0.48	<10	8	-	1090	+++	C3 cap	+++	[71]
2	12/M	DDD	AKI on CGN	+++	30–49	0.44	31.1	48	27	333	+++	C3 cap	+++	[70]
3	6/F	DDD	Asymptomatic	±	20–29	0.36	23.5	36	19	290	++	C3 cap	+	our case
4	62/F	MPGN I	AKI on CGN	+++	20–29	3.6	21.7	-	-	170	++	IgG, IgM, C3 cap	++	[73]
5	6/F	HSPN	Abd pain, purpura	+++	many	0.35	67.1	89	32	154	++	IgA, C3 mes-cap	++	our case
6	25/M	HSPN	Abd pain, purpura	+	30–49	1.3	28	50	22	743	+++	IgA, C3 mes-cap	+++	[76]
7	71/M	SVV	Pain and swelling of the rt. ankle	+++	50–99	3.28	-	42	27	512	+++	C3 mes-cap	+++	[75]
8	17/M	GPA	AGN	++	100<	7.53	-	10	26	301	−	C3 mes	+	[74]

SIRN = streptococcal infection-related nephritis; UP = urine protein; U-RBC = urine red blood cells/high power field; SCr = serum creatinine; CH50 = total hemolytic complement; ASO = anti-streptolysin O; EP = endocapillary proliferation; IF = immunofluorescence; NAPlr = nephritis-associated plasmin receptor; DDD = dense deposit disease; AGN = acute glomerulonephritis; cap = capillary; mes = mesangial; CGN = chronic glomerulonephritis; MPGN = membranoproliferative glomerulonephritis; AKI = acute kidney injury; HSPN = Henoch Schönlein purpura nephritis; Abd = abdominal; SVV = small vessel vasculitis; GPA = granulomatosis with polyangiitis; IgA-IRGN = IgA-dominant IRGN.

**Table 6 ijms-23-09974-t006:** Clinical and pathological profiles in patients with NAPlr-positive non-streptococcal IRGN.

Case	Age/ Gender	Infection Focus	Pathogen	Complement C3/C4/CH50	NAPlr /PA	Light Microscopy	IF	EM (Dense Deposits)	Ref.
1	12/F	respiratory infection	*Streptococcus* *pneumoniae*	10/33/-	+/+	cellular crescent formation, endocapillary proliferative GN	C3	GBM hump (−)	[77]
2	64/M	endocarditis	*Aggregatibacter* *actinomycetemcomitans*	86/16/-	+/−	proliferative GN with inflammatory cell infiltration	IgG IgM C3 C1q IgA	Subend Mes hump (−)	[78]
3	7/F	respiratory infection	*Mycoplasma pneumoniae*	WNL	+/+	endocapillary proliferation and cellular crescents	IgA IgM C3	Mes Parames Subendo hump (−)	[79]
4	70/M	cellulitis	*Staphylococcus* *aureus (MSSA)*	118/43.9/75.3	+/+	endocapillary proliferative GN	IgG IgA IgM C3 C1q	GBM Parames hump (−)	Our case

IF = immunofluorescence; EM = electron microscopy;
PA= plasmin activity; GN = glomerulonephritis; GBM = glomerular basement membrane; Subend = subendothelium; Mes = mesangium; WNL = within normal limits; Parames = paramesangium; ANCA = antineutrophilic cytoplasmic antibody.

**Table 7 ijms-23-09974-t007:** Identity, similarity, and C-terminal amino acid sequences of bacterial GAPDH in NAPlr-positive patients with non-streptococcal IRGN.

Pathogen	Amino Acid Sequence of GAPDH			
	Total Amino Acid	Identity Similarity (%)		C Terminal Amino Acid
*Streptococcal pyogenes*	336	-	-	T L E Y F A K I A K
*Streptococcus pneumoniae*	359	92	99	T L E Y F A K I A K
*Aggregatibacter* *actinomycetemcomitans*	334	50	85	L V A H V Y N Y K D
*Mycoplasma pneumoniae*	337	54	87	V R V V N Y C A K L
*Staphylococcus aureus (MSSA)*	336	67	92	T L A Y L A E L S K

**Table 8 ijms-23-09974-t008:** Clinical and pathological profiles in patients with IRGN induced by either no or unknown pathogens who were positive for NAPlr and plasmin activity.

Case	Age/ Gender	Infection Focus	Underlying Disease	Pathogen	Complement C3/C4/CH50	NAPlr/ PA	Light Microscopy	IF	EM (Dense Deposits)	Ref.
5	68/M	sinusitis	-	unknown	25.1/20.8/<12	+/+	proliferative and exudative GN, MPGN type I	IgG C3	Parames hump (−)	[89]
6	17/F	unknown	C3G	unknown	6/15/<12	+/+	lobulation of the glomerular capillary tufts, proliferation of glomerular mesangial cells, increased mesangial matrix, MPGN type III	C3	Subendo Subepi	[85]
7	55/M	unknown	CFH mutation	unknown	108/43/61.6	+/+	cellular crescents and diffuse endocapillary and mesangial proliferation with double contour formation along the capillary walls	C3 IgG	Subepi Subendo hump (−)	[83]
8	70/M	respiratory infection	-	unknown	80/11.8/15.8	+/+	endocapillary proliferative GN	IgG C3	Subepi hump (+)	[84]
9	75/M	unknown	ANCA negative SVV	unknown	105/26/52.7	+/+	cellular crescents/segmental endocapillary proliferation with neutrophils and tuft necrosis	C3	Mes, GBM, Subepi, hump (+)	[86]
10	82/M	unknown	-	unknown	64/25/-	+/+	subendothelial deposits including wire loop lesions, endocapillary proliferation, cellular crescents	IgA C3 IgG/IgM C1q/C4	Subendo, Mes hump (−)	[88]
11	27/M	unknown	IgA nephropathy	unknown	18/23/-	+/+	mesangial cell proliferation, increased mesangial matrix increasing, endocapillary proliferation	IgA C3	Mes hump (−)	[87]

PA = plasmin activity; IF = immunofluorescence; EM = electron microscopy; GN = glomerulonephritis; MPGN = membranoproliferative glomerulonephritis; Parames = paramesangium; C3G = C3 glomerulopathy; Subendo = subendothelium; Subepi = subepithelium; CFH = complement factor H; Mes = mesangium; WNL = within normal limits; ANCA = antineutrophilic cytoplasmic antibody; SVV = small vessel vasculitis; Mes = mesangium; GBM = glomerular basement membrane.

## Data Availability

All data generated in our laboratory is available to other investigators upon request.

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
