# Peer review of "Nephritis-Associated Plasmin Receptor (NAPlr): An Essential Inducer of C3-Dominant Glomerular Injury and a Potential Key Diagnostic Biomarker of Infection-Related Glomerulonephritis (IRGN)"

_ijms, 2022, doi:10.3390/ijms23179974_

Round 1
Reviewer 1 Report
Dear Authors, The findings of your laboratory and other research data are appropriately used to provide a clear explanation for the development of IRGN. The pathogenesis of IC-, or C3-dominant GN explain in an easy-to-understand manner, and the content is generally reasonable. Also, you showed that the analysis of NAPlr is useful as a diagnostic and therapeutic indicator for IRGN other than PSAGN, even for latent or deep-seated infections. This point is highly valued as a particularly important point in this review.

Reviewer 2 Report
Yoshizawa and coauthors propose the role of nephritis-associated plasmin receptor (NAPlr) in the pathogenesis of infection-related glomerulonephritis (IRGN). This is a logical continuation of the previous works by the same group who discovered the NAPlr. Even though this is an interesting review, there are some concerns about the proposals:
1. Whereas the authors hypothesize the role of NAPlr in the IRGN associated with Streptococcal infection, its role in other IRGN, such as Staphylococcal, is questionable. Findings in the kidney biopsy in patients with Staphylococcal IRGN usually show proliferative glomerulonephritis with IgA and C3 immune complexes in the glomeruli, not IgG, and the “humps” are uncommon in such cases (Nat Rev Nephrol. 2020 Jan;16(1):32-50; Clin J Am Soc Nephrol. 2017 Jan 6;12(1):39-49.). There is no sufficient data that NAPlr is present in patients with Staphylococcal IRGN. This limitation should be emphasized in the review and in the title.
2. The deposition of immune complexes in many diseases does not correspond with inflammatory lesions in the glomeruli (e.g. IgA nephropathy) and, in some cases, there is no immune complexes in active inflammatory process (e.g. pauci-immune ANCA-associated GN). Therefore, the hypothesis that localization of C3 and IgG containing deposits in the epithelial site of the GBM is not associated with inflammation on the endothelial site of the GBM is not well supported by data.
3. Figure 9 in the current manuscript resembles Figure 1 from the previous work by the same group (Int J Mol Sci. 2020 Apr 8;21(7):2595). Even though the figure is not identical, the similarity is obvious.
4. Diagnostic criteria that are described for IRGN (lines 460-466) are not very accurate. In renal pathology, there is no strict criterion to make the diagnosis, only constellation of clinical history, laboratory data and morphologic findings allows making the correct diagnosis. Indeed, it is difficult to distinguish on the morphologic level C3 GN and post0infections glomerulonephritis, but differential diagnosis with other diseases usually is not very difficult. BTW, MPGN is rarely diagnosed by pathologists nowadays, most of the cases represent different forms of C3 GN.
Reviewer 3 Report
Very interesting, stimulating, and complete review.
The educational objective of the review is achieved especially for the nephrologist.
May be some paragraphs are excessively extensive
I think figure 15 is not necessary (or suppt data)
A list of abbreviations is necessary.
tables and figures are very informative and easy to read.
Reviewer 4 Report
This review article discusses the role of nephritis-associated plasmin receptor (NAPlr) in C3-dominant glomerular injury and in diagnosis of infection-related glomerulonephritis.
This topic is new and of interest. However, in my opinion the conclusions are not clearly enough defined. The authors should discuss what are the key findings and weaknesses in the research done in this field so far?
Round 2
Reviewer 2 Report
the authors address all questions.